# The Soybean High Density ‘Forrest’ by ‘Williams 82’ SNP-Based Genetic Linkage Map Identifies QTL and Candidate Genes for Seed Isoflavone Content

**DOI:** 10.3390/plants10102029

**Published:** 2021-09-27

**Authors:** Dounya Knizia, Jiazheng Yuan, Nacer Bellaloui, Tri Vuong, Mariola Usovsky, Qijian Song, Frances Betts, Teresa Register, Earl Williams, Naoufal Lakhssassi, Hamid Mazouz, Henry T. Nguyen, Khalid Meksem, Alemu Mengistu, My Abdelmajid Kassem

**Affiliations:** 1Department of Plant, Soil, and Agricultural Systems, Southern Illinois University, Carbondale, IL 62901, USA; dounya.knizia@siu.edu (D.K.); naoufal.lakhssassi@siu.edu (N.L.); meksem@siu.edu (K.M.); 2Laboratoire de Biotechnologies & Valorisation des Bio-Ressources (BioVar), Department de Biology, Faculté des Sciences, Université Moulay Ismail, Meknès 50000, Morocco; H.MAZOUZ@fs-umi.ac.ma; 3Plant Genomics and Biotechnology Laboratory, Department of Biological and Forensic Sciences, Fayetteville State University, Fayetteville, NC 28301, USA; jyuan@uncfsu.edu (J.Y.); fbetts@broncos.uncfsu.edu (F.B.); tregist2@broncos.uncfsu.edu (T.R.); ewilli17@broncos.uncfsu.edu (E.W.); 4Crop Genetics Research Unit, USDA, Agriculture Research Service, 141 Experiment Station Road, Stoneville, MS 38776, USA; nacer.bellaloui@usda.gov; 5Division of Plant Science and Technology, University of Missouri, Columbia, MO 65211, USA; vuongt@missouri.edu (T.V.); klepadlom@missouri.edu (M.U.); nguyenhenry@missouri.edu (H.T.N.); 6Soybean Genomics and Improvement Laboratory, USDA-ARS, Beltsville, MD 20705, USA; qijian.song@usda.gov; 7Crop Genetics Research Unit, USDA, Agricultural Research Service, Jackson, TN 38301, USA; alemu.mengistu@usda.gov

**Keywords:** soybean, RIL, Forrest, Williams 82, linkage map, isoflavone, daidzein, genistein, glycitein, SNP

## Abstract

Isoflavones are secondary metabolites that are abundant in soybean and other legume seeds providing health and nutrition benefits for both humans and animals. The objectives of this study were to construct a single nucleotide polymorphism (SNP)-based genetic linkage map using the ‘Forrest’ by ‘Williams 82’ (F×W82) recombinant inbred line (RIL) population (*n* = 306); map quantitative trait loci (QTL) for seed daidzein, genistein, glycitein, and total isoflavone contents in two environments over two years (NC-2018 and IL-2020); identify candidate genes for seed isoflavone. The FXW82 SNP-based map was composed of 2075 SNPs and covered 4029.9 cM. A total of 27 QTL that control various seed isoflavone traits have been identified and mapped on chromosomes (Chrs.) 2, 4, 5, 6, 10, 12, 15, 19, and 20 in both NC-2018 (13 QTL) and IL-2020 (14 QTL). The six QTL regions on Chrs. 2, 4, 5, 12, 15, and 19 are novel regions while the other 21 QTL have been identified by other studies using different biparental mapping populations or genome-wide association studies (GWAS). A total of 130 candidate genes involved in isoflavone biosynthetic pathways have been identified on all 20 Chrs. And among them 16 have been identified and located within or close to the QTL identified in this study. Moreover, transcripts from four genes (*Glyma.10G058200*, *Glyma.06G143000*, *Glyma.06G137100*, and *Glyma.06G137300*) were highly abundant in Forrest and Williams 82 seeds. The identified QTL and four candidate genes will be useful in breeding programs to develop soybean cultivars with high beneficial isoflavone contents.

## 1. Introduction

Soybean seeds are rich in secondary metabolites beneficial for human and animal consumption including tocopherols, phenolic compounds, saponins, and isoflavones such as genistein, daidzein, and glycitein that showed beneficial health and nutrition effects in animals and humans [1,2,3]. It is well established that isoflavones reduce menopausal symptoms, low density lipoprotein (LDL) cholesterol levels, breast and prostate cancers risks, improve the immune system [4,5,6,7,8,9,10,11], and play an important role in nitrogen fixation and defense against pathogens [12].

Due to these benefits and others, isoflavones, especially genistein, daidzein, and glycitein, have been widely studied during the past decades [13,14] and many studies tried to genetically map quantitative trait loci (QTL) that control seed genistein, daidzein, glycitein, and total isoflavone content as well as their precursors such as daidzin, glycitin, genistin, malonyldaidzin, malonylglycitin, malonylgenistin, etc., using different molecular markers such as AFLPs, RFLPs, SSRs, SNPs [15,16,17,18,19,20,21,22,23,24,25,26,27]. For example, using the ‘Essex’ by ‘Forrest’ recombinant inbred line (RIL) population (*n* = 100) and 250+ simple sequence repeat (SSR) markers, 11 QTL that control genistein, daidzein, glycitein, and total isoflavone contents have been identified on Chrs. 1, 3, 7, 8, 11, and 18 [15,16]. Likewise, Liang et al. (2010) used the ‘Jindou 23’ by ‘Huibuzhi’ RIL population (*n* = 474) and identified six QTL that control isoflavone contents and mapped them on soybean Chrs. 3, 16, 17, and 18 [18]. In another study, Smallwood et al. (2014) identified 3, 5, 7, and 6 QTL that control seed glycitein, daidzein, genistein, and total isoflavone contents, respectively [20]. Using the ‘Zhongdou 27’ by ‘Jiunong 20’ RIL population (*n* = 130) and 194 SSR markers, Han et al. (2015), identified 6, 5, 3, and 7 QTL that control seed glycitein, daidzein, genistein, and total isoflavone contents, respectively [24]. Akond et al. (2015) used the ‘Hamilton’ by ‘Spencer’ RIL population (*n* = 93), genotyped it with 1502 SNPs, and identified a major QTL that controls both seed daidzein and total isoflavone contents on Chr. 6 and a minor QTL that controls seed glycitein content on Chr. 18 [22]. Recently, the authors of [23] used ‘Aokimame’ by ‘Fukuyutaka’ and ‘Kumaji-1’ by ‘Fukuyutaka’ RIL populations and identified one QTL that controls malonylgenistin on Chr. 12 and two QTL that control malonylglycitin on Chrs. 11 and 15 [23]. Besides using biparental mapping populations, other researchers used natural populations and genome wide association studies (GWAS) to map QTL that control seed isoflavone contents and identified candidate genes within these QTL regions [28,29,30,31,32].

The objectives of this study were to construct a SNP-based genetic linkage map using the F×W82 RIL population (*n* = 306); map quantitative trait loci (QTL) for seed daidzein, genistein, glycitein, and total isoflavone contents in two environments over two years; identify candidate genes involved in soybean seed isoflavone biosynthesis.

## 2. Results and Discussion

### 2.1. The SNP-Based Genetic Map

A total of 5405 SNP markers were generated from the Infinium SNP6K BeadChips-based genotyping among 306 RILs, from which 2075 polymorphic SNPs were mapped on the 20 soybean chromosomes (Table 1, Figure 1). The F×W82 genetic map covered 4029.9 cM with an average marker density of 1.94 cM (Table 1). The genetic length ranged from 153.7 cM for Chr. 18 to 308.3 cM for Chr. 2 (Table 1). The polymorphism of SNPs in this RIL population (38.4%), number of linked SNPs, and map coverage were comparable to other reported SNP-based genetic linkage maps of soybean [33,34]. For example, in Akond et al. (2013) [33], only 27.33% of SNPs (1465/5361 × 100) have been used to construct the genetic map based on excluding missing data (~20%) and heterozygosity (3.99%). Polymorphic markers between parents (MD96-5722 and Spencer) among the 1465 SNPs used was 44.8% (657/1465 × 100) [33]. Likewise, in Kassem et al. (2012) [34], polymorphic markers between parents (PI 438489B and Hamilton) among the 1465 SNPs used was 44.2% (657/1465 × 100) [34].

### 2.2. Isoflavone Contents Frequency Distribution, Heritability, and Correlation

The seed isoflavone contents were normally distributed in the FxW82 RIL population based on Shapiro–Wilk’s method for normality test, even though the positive or negative skewness and kurtosis value (>3) were observed in the RIL population (Table 2; Figure 1). The individual component of isoflavone also displayed small ranges of phenotypic variations in the seeds obtained from two geographically diverse field trials (Table 2, Figure 2). Daidzein 2018 in Spring Lake, NC had the highest coefficients of variation (CV) value (19.37%); however, the CV of this trait in Carbondale, IL (2020) was only 12.59% suggesting that phenotypic variability among isoflavone contents was impacted by different environmental conditions.

Azam et al. (2020) [35] reported that the total isoflavones ranged from 745 to 5253.98 μg/g, with highest mean of 2689.27 μg/g observed in some regions and up to 2518.91 and 1942.78 μg/g in others due to climatic conditions. Similar results have been reported by other studies [25,26,27,36,37,38]. Our results showed over 1000 μg/g and in some cases over 1100 μg/g. Therefore, the total concentrations of isoflavones are in the expected range of soybean seed. In addition, there is no premium to be given to growers for soybean seed isoflavone content and no docking is done at the grain elevator for seed isoflavone. Isoflavone concentrations vary depending on the year and environmental growing conditions. Although isoflavones are genetically controlled, environmental conditions including temperature, drought, absence or presence of diseases each year, many other biotic and abiotic factors can significantly affect the contents (by increasing or decreasing) and profile of isoflavone.

The broad sense heritability of percentage dry weight for daidzein, glycitein, and genistein across two different environments over two years appeared to be quite different. Glycitein had the highest heritability (72.4%) and the values for both daidzein and genistein were 42.8% and 42.5%, respectively, which displayed a similar fashion. The lower heritability of daidzein and genistein contents suggested that some portion of phenotypic variation was still not detected by the mapped QTL due to the complexity of these traits. The genotype–environment interactions still played a significant role in the molecular formation of daidzein and genistein molecules in soybean seeds based on our two-way ANOVA analysis because the σ_GE2_ is relatively high compared to that of glycitein (data not shown). It will certainly impact future breeding strategies for trait improvement based on the data we presented on these traits.

We used type I sum of squares (ANOVA (model)) function in R program to obtain the Sum Sq and Mean Sq and calculated σ_G_^2^ and σ_GE_^2^ for each trait (Table 3). However, σ_e_^2^ was 0 due to limited replicates. In this study, we only had three technical replicates due to cost effect of this student-centered project, but these replicates could only be considered as one biological replicate and hence, F value and probability could not be generated (Table 3). The FxW82 RIL population derived from two parental cultivars with different maturity groups (MGs). Forrest belongs to MG5-6 and Williams 82 to MG2-3 suggesting that the locations may play an important role on major agronomic traits including seed isoflavone. Based on our data (Figure 2), glycitein showed less correlation with daidzein and genistein which may indicate that its production may be less impacted by environment. Furthermore, Fayetteville, NC is a subtropic favorable weather for MG6-7 soybeans while Carbondale, IL is the favorable weather for MG 4-5. Therefore, further studies of the seed isoflavone in the FxW82 RIL population in different environments would be beneficial.

The correlogram demonstrates a novel correlation among these assessed traits (Figure 2). Based on the unassorted data (all lines were included), each of the isoflavone components was positively correlated with the other sister isoflavones (*p* < 0.001) from the same geographical location but negatively correlated with the isoflavones from the other location inferring that the production of these isoflavones has been strongly impacted both by genotype and environmental conditions. The assorted data (lines tested in both locations) showed similar positivity, but the level of negative correlation was low (Figure 2). To the best of our knowledge, this observation has not been described in other studies.

### 2.3. Seed Isoflavone Contents QTL

Both interval mapping (IM) and composite interval mapping (CIM) methods of WinQTL Cartographer 2.5 [39] were used to identify QTL for seed daidzein, genistein, glycitein, and total isoflavone contents in the present RIL population. A total of 27 QTL that control seed isoflavone contents have been identified in this population in both NC-2018 (13 QTL) and IL-2020 (14 QTL) (Table 4, Figure 3 and Appendix A).

In Carbondale, IL (IL-2020), one QTL that controls seed daidzein content (*qDAID03*) has been identified and mapped on Chr. 2 and one QTL that controls seed genistein content (*qGEN04*) has been identified and mapped on Chr. 4 (Table 4, Figure 3 and Appendix A). One QTL that controls seed daidzein content (*qDAID04*) and two QTL that control seed genistein content (*qGEN01* and *qGEN05*) have been identified and mapped on Chr. 10. Two QTL that control each of seed daidzein (*qDAID01* and *qDAID05*) and seed genistein contents (*qGEN02* and *qGEN06*) have been identified and mapped on Chr. 12 (Table 4, Figure 3 and Appendix A). One QTL that controls seed glycitein (qGLY01) has been identified and mapped on Chr. 15 (Table 4, Figure 3 and Appendix A). Two QTL that control each of seed daidzein (*qDAID02* and *qDAID06*) and genistein contents (*qGEN03* and *qGEN07*) have been identified and mapped on Chr. 20 (Table 4, Figure 3 and Appendix A).

In Spring Lake, NC (NC-2020), one QTL that controls each of seed daidzein (*qDAID01*), genistein (*qGEN03*), and glycitein contents (*qGLY02*) have been identified and mapped on Chr. 5 (Table 4, Figure 3 and Appendix A). Two QTL that control seed genistein content (*qGEN01* and *qGEN04*) and one QTL that controls seed daidzein content (*qDAID02*) have been identified and mapped on Chr. 6 (Table 4, Figure 3 and Appendix A). Two QTL that control each of seed genistein (*qGEN02* and *qGEN05*) and glycitein contents (*qGLY01* and *qGLY03*) have been identified and mapped on Chr. 12 (Table 4, Figure 3 and Appendix A). One QTL that controls each of seed daidzein (*qDAID03*) and glycitein contents (*qGLY04*) have been identified and mapped on Chr. 19 (Table 4, Figure 3 and Appendix A). One QTL that controls seed genistein content (*qGEN06*) has been identified and mapped on Chr. 20 (Table 4, Figure 3 and Appendix A). No QTL that controls total seed isoflavone contents has been identified in both years and locations.

No previous studies identified QTL that control seed isoflavone contents in the QTL region identified on Chr. 2 (*qDAID03*-(IL-2020), 23–25 cM), indicating that this is a novel QTL region; however, other studies identified QTL that control seed calcium content, plant height, and few other traits [40,41]. Likewise, no other studies identified QTL that control seed isoflavone contents in the QTL region identified on Chr. 4 (*qGEN04*-(IL-2020), 189.3–191.3 cM) which indicates that it is also a novel QTL region. The length of Chr. 4 in the soybean consensus map is only 136 cM [29,31]. Additionally, no other studies identified QTL that control seed isoflavone contents in the QTL region identified on Chr. 5 (*qDAID01*-(NC-2018), *qGEN03*-(NC-2018), and *qGLY02*-(NC-2018), 152.4–166.4 cM) which indicates the discovery of a novel QTL region. The length of Chr. 5 in the soybean consensus map is only 104 cM [29,31]. Interestingly, within the same QTL region that controls seed genistein and daidzein contents on Chr. 6 (*q-GEN01*-(NC-2018), *qGEN04*-(NC-2018), and *qDAID02*-(NC-2018), other studies identified QTL that control seed genistein, daidzein, glycitein, and total isoflavone contents (see a summary in [30]) which is coherent with our results making it an important genomic region to further investigate for candidate genes. Other studies identified QTL that control seed protein, oil, γ–tocopherol, and amino acids contents, and few other traits [29,31]. Interestingly, within the same QTL region that controls seed genistein and daidzein contents on Chr. 10 (*q-GEN01*-(IL-2020), *qGEN05*-(IL-2020), and *qDAID04*-(IL-2020), 128.8–132.9 cM), another study identified QTL that control seed isoflavone content [41,42] which is consistent with our data making it an important genomic region for gene discovery. In fact, two candidate genes have been previously identified in this region [42,43]. Two QTL regions have been identified on Chr. 12. The first region containing QTL that control seed genistein and daidzein contents (*qGEN02*-(IL-2020), *qGEN06*-(IL-2020), and *qDAID05*-(IL-2020), 51.2–71.9 cM). Interestingly, other studies identified QTL that control seed daidzein, genistein, glycitein, and total isoflavone contents in the same QTL region (see a summary in [30]) which makes it an important genomic region for discovering novel candidate genes. The second region contained QTL that control seed genistein, and glycitein contents (*qGEN02*-(NC-2018), *qGEN05*-(NC-2018), *qGLY01*-(NC-2018), and *qGLY03*-(NC-2018), 169.3–181.3 cM). No other studies identified QTL that control seed isoflavone contents in this second QTL region which indicates that it is also a novel QTL region. The length of Chr. 12 in the soybean consensus map is only 125 cM and the second QTL region identified here falls outside of its current limit [29,31]. No previous studies identified QTL that control seed isoflavone contents in the QTL region identified on Chr. 15 (*qGLY01*-(IL-2020), 212.3–218.8 cM) which indicates that it is a novel QTL region. The length of Chr. 15 in the soybean consensus map is only 85 cM and the QTL region identified here falls outside of its current limit [29,31]. Two QTL regions have been identified on Chr. 19. The first region contains QTL that control seed glycitein content (*qGLY04*-(NC-2018), 34.3–36.3 cM). Interestingly, other studies identified QTL that control seed genistein, daidzein, and isoflavone content within the same QTL region (see a summary in [30]). Previous studies identified also QTL for seed protein content (see a summary in [44]). The second region contained QTL that control (*qDAID03*-(NC-2018), 108.5–110.5 cM). No other studies identified QTL that control seed isoflavone contents in this QTL region, making it a novel QTL region. Two QTL regions have been identified on Chr. 20. Within the first region containing QTL that control seed glycitein content (*qGEN05*-(NC-2018), 0–2 cM), other studies identified QTL that control seed daidzein (*qD20*), genistein (*qG20*), malonyldaidzein (*qMD20*), malonylgenistein (*qMG20*), and total isoflavone content (*qTIF20*) [41,44] which makes it an important region to investigate further for candidate genes. In addition, other studies identified QTL for seed calcium [30,44] and sucrose contents within this QTL region as well (see a summary in [44]). Within the second region containing QTL that control seed daidzein and genistein contents (*qDAID02*-(IL-2020), *qDAID06*-(IL-2020), *qGEN03*-(IL-2020), and *qGEN07*-(IL-2020), 64.3–80.5 cM), other studies identified QTL that control seed genistein content (*qGEN20*, [17]) and seed daidzein and glycitein contents (*qGC|proI_1 and qDZ|proI_2*, [39,41] which makes it another important region to investigate further for candidate genes. In addition, other studies identified QTL that control seed phytate, stearic acid, calcium, alpha-tocopherol, and few amino acids [44].

### 2.4. Seed Isoflavone Candidate Genes

A total of 130 candidate genes involved in soybean isoflavone biosynthetic pathway have been identified in all 20 soybean Chrs. (Appendix A); however, 16 candidate genes have been identified within or close to the seed isoflavone QTL identified in this study on Chrs. 2, 6, 10, 12, 15, 19, and 20 (Figure 4, Table 5).

Among them, the 4′-methoxyisoflavone 2-hydroxylase gene (*Glyma.02G067900*) and the chalcone synthase gene (*Glyma.02G130400*) are located at 3.7 and 11.11 cM, respectively, from qDAID03-(NC-2018) on Chr. 2 (Figure 4, Table 5 and Appendix A). *Glyma.06G128200* is a flavonol synthase gene located at 5.52 cM from qGEN01-(NC-2018), qDAID02-(NC-2018), and qGEN04-(NC-2018) on Chr. 6. The flavonol 3-O-methyltransferase genes (*Glyma.06G137100* and *Glyma.06G137300*) as well as the chalcone-flavonone isomerase gene (*Glyma.06G143000*) are located at 6 cM from qGEN01-(NC-2018) on Chr. 6 (Figure 4, Table 5 and Appendix A). *Glyma.10G058200* is a phenylalanine ammonia-lyase gene that is located 0.6 cM from qGEN01-(IL-2020), qDAID04-(IL-2020) and qGEN05-(IL-2020) on Chr. 10 (Figure 4, Table 4 and Appendix A). *Glyma.12G067000* and *Glyma.12G067100* are located within qDAID01-(IL-2020), qGEN02-(IL-2020) and qDAID05-(IL-2020) on Chr. 12 and near to (<4 cM) qGEN02-(NC-2018), qGLY01-(NC-2018), qGEN05-(NC-2018), qGEN06-(IL-2020) and qGLY03-(NC-2018) on Chr. 12 (Figure 4, Table 5 and Appendix A). Both genes are Cytochrome P450 CYP2 subfamily genes; *Glyma.12G067000* was classified as an isoflavone synthase II gene and *Glyma.12G067100* as its duplicate with 95% identical nucleotide positions in the protein coding sequence ([42] Fliegmann et al., 2010). The trans-feruloyl-CoA synthase gene *Glyma.15G001700* is located at 0.56 cM from qGLY01-(IL-2020) on Chr. 15. The Isoflavone 3′-hydroxylase gene *Glyma.15G156300* and the Isoflavone 2′-hydroxylase gene *Glyma.15G156100* are located at about 11 cM from qGLY01-(IL-2020) on Chr. 15 (Figure 4, Table 5 and Appendix A). *Glyma.20G027800* is an isoflavone reductase gene that is located within qDAID06-(IL-2020) and at 0.62 cM from qDAID02-(IL-2020), qGEN03-(IL-2020), and qGEN06-(NC-2018) and 2.44 cM from qGEN07-(IL-2020) on Chr. 20 (Figure 4, Table 5 and Appendix A). Three genes *Glyma.19G030500*, *Glyma.19G030700*, and *Glyma.19G030800* encoding for an isoflavone 7-O-glucoside-6″-O-malonyltransferase gene family are located at less than 1 cM from qDAID03-(NC-2018) and qGLY04-(NC-2018) on Chr. 19 (Figure 4, Table 5 and Appendix A). Interestingly, Wu et al. (2020) identified seven candidate genes including the mitogen-activated protein kinase (MPK) gene (*Glyma.08G309500*) within the seed isoflavone QTL identified on Chr. 8 [29,31]. A summary of seed isoflavone QTL and corresponding candidate genes for over two decades of research (1999–2020) can be found in Kassem [30]. Recently, Yang et al. (2021) [32] identified four candidate genes including GSTT1a (Glyma.05G206900), GSTT1b (Glyma.05G207000), and the transcription factor (TF) GL3 (Glyma.05G208300) on Chr. 5, and GSTL3 (Glyma.13G135600) on Chr. 13 [32].

### 2.5. Expression Analysis

To gain insight into the role of isoflavone genes in soybean seeds, RNA-Seq analysis was conducted to check the expression levels of the 16 candidate genes that are located within or near the isoflavone QTL identified in FxW82 RIL population. Expression analysis of these genes showed that four genes, Glyma.10G058200, Glyma.06G143000, Glyma.06G137100, and Glyma.06G137300, are highly expressed in seeds of both Forrest and Williams 82 cultivars. Whereas, Glyma.19G030800 is highly expressed in Williams 82 and have a low expression in Forrest cv.; the rest of the 16 genes showed lower expressions; whereas Glyma.02G067900 and Glyma.15G156300 were not expressed neither in Forrest nor in Williams 82 cultivars (Figure 5).

Surprisingly, Glyma.10G058200 expression in Forrest cv. is higher than its expression in Williams 82. This could explain the presence of RILs from the FxWI cross that showed higher daidzein, glycitein or genistein content than the parent Williams 82 (parent with the high isoflavones content), these lines inherited most likely the beneficial alleles from both parents Forrest and Williams 82.

### 2.6. Conclusions

In conclusion, we constructed the FxW82 dense SNP-based genetic linkage map (2075 SNPs and 4029.9 cM covered) and identified 27 QTL that control soybean seed isoflavone contents and 16 candidate genes involved in soybean isoflavone biosynthetic pathways among which four candidate genes are highly expressed in seeds of both Forrest and Williams 82, in addition to Glyma.19G030800 that has a higher expression profile in Williams 82 compared to its expression in Forrest cv. (Figure 5).

A comparison of the Forrest and Williams 82 sequences of these four genes has shown that two of these genes have SNPs between Forrest and Williams 82 sequences, Glyma.10G058200 and Glyma.06G143000. Glyma.10G058200 has ten SNPs, one SNP is upstream 5′ UTR, four SNPs are located at the intron, two SNPs are at the exon 1, one of them caused a missense mutation (A127G) and the other one caused a silent mutation (A32A). The last three are in the 3′ UTR downstream region. For Glyma.06G143000, there is only one SNP located in the 5‘UTR upstream region (Figure 6). These SNPs could potentially play a role in the difference of isoflavones content between Forrest and Williams 82 cultivars. Moreover, Glyma.10G058200 and Glyma.06G143000 are highly expressed in the seed tissue of both Forrest and Williams 82 (Figure 5). Glyma.10G058200 is associated with qGEN01-(IL-2020) QTL, qDAID04-(IL-2020) QTL and qGEN05-(IL-2020) QTL. Additionally, Glyma.06G143000 is associated with qGEN01-(NC-2018) QTL. The two genes could be useful for breeding for increased isoflavones content in soybeans.

## 3. Materials and Methods

### 3.1. Plant Material and Growth Conditions

In this study, we used ‘Forrest’ × ‘Williams 82’ RIL population (*n* = 306). The cultivar ‘Forrest’ was derived from the cross of ‘Dyer’ and ‘Bragg’ developed by USDA [46]. The cultivar ‘Williams 82’ was derived from the cross of ‘Williams’ and ‘Kingwa’ [47]. The genomes of soybean cultivars including Forrest and Williams 82 genomes are duplicated polyploid genomes with highly conserved gene-rich regions [48]. Originally, the ‘Forrest’ × ‘Williams 82’ RIL population was developed with more than 1000 RILs [49]. The genetic map used in this study was based on 306 RILs and 2075 SNP markers; however, QTL data analysis in Spring Lake, NC-2018 was based on 190 RILs.

The RIL population was evaluated in a farm in Spring Lake, NC (35.17° N, 78.97° W) in 2018 and in a farm in Carbondale, IL (37° N, 89° W) in 2020. Seeds of parents (Forrest and Williams 82) were sown directly in the field in a randomized complete block design (RCB) and 75 cm row spacing between seeds with three replicates. The plants were watered by drip irrigation and kept in the field until maturity. No pesticide, herbicide, or fertilizer were applied. In September 2018, hurricane Florence hit NC and its winds of 90+ mph damaged the fence in the farm in Spring Lake, NC and the deer damaged about 119 RILs; therefore, QTL data analysis for this location involved 187 RILs (*n* = 187). The plants grown in Carbondale, IL (*n* = 306) were not damaged.

In Spring Lake, NC (2018) during the growing season (May–Sept.), the temperatures ranged from 7.2 to 35 °C, it was partly (40%) to mostly cloudy (80%), wind speeds ranging from 55 to 90+ mph (hurricane Florence), and humidity comfort level ranged from comfortable to miserable [50]. The soil type in this location is mainly sandy (NC Sandhills). In Carbondale, IL (2020), the temperatures ranged from 7.2 to 29.4 °C, it was mostly clear (25%) to mostly cloudy (80%), wind speeds ranging from 30 to 38 mph, and humidity comfort level ranged from comfortable to miserable (weatherspark.com). The field was treated first using Firestorm (contains Paraquat dichloride) to control annual grass and broad-leaved weeds. As pre-emergent herbicide, Dual II Magnum Herbicide with long-lasting control of most annual grasses and small-seeded broadleaf weeds was used to eliminate early-season weed competition. As post-emergent herbicide, Round Up Pro Concentrate (50.2% Glyphosate) was used/sprayed between the rows to control emerging weed. Weed grown inside the plastic mulch very close to the soybeans were removed manually. The soil type in this location is mainly silty clay loam (Southern IL).

### 3.2. Isoflavone Quantification

Mature seeds of parents Forrest and Williams 82, and the 190 RILs were analyzed for the concentrations aglycones daidzein, genistein, and glycitein. Approximately 25 g of mature seeds from each plot were ground using a Laboratory Mill 3600 (Perten, Springfield, IL, USA). Concentrations of daidzein, genistein, and glycitein were analyzed using a near-infrared reflectance (NIR) diode array feed analyzer (Perten, Spring Field, IL, USA). The calibration equation has been updated every 6 months to 1 year and developed using the Thermo Galactic Grams PLS IQ software developed by Perten Company (Perten, Springfield, IL, USA). Thermo Galactic Grams PLS IQ from Perten (Perten) was used to develop the calibration equations, which was initially developed by the University of Minnesota. Descriptions of quantifying daidzein, genistein, glycitein and total isoflavone contents was reported by others (Akond et al., 2015 [22]; Bellaloui et al., 2012 [51]; Wang et al., 2019 [38]). The calibration equation development and updating for isoflavones was based on standard laboratory analytical methods (AOAC 2002) using High Performance Liquid Chromatography (HPLC) and use of adequate number of samples, providing sufficiently accurate estimations of isoflavones concentrations. The produced calibration equation was characterized by high correlation, indicating the accuracy of the method. The concentrations were calculated on a seed dry matter basis.

### 3.3. DNA Isolation, SNP Genotyping, and Genetic Map Construction

Genomic DNA of the RIL population and the parents were extracted using a standard cetyltrimethyl ammonium bromide (CTAB) method with minor modifications as previously described [52]. DNA concentration was quantified with a spectrophotometer (NanoDrop Technologies Inc., Centreville, DE, USA) and then normalized at 50 ng/µL for genotyping. SNP genotyping was performed in the Soybean Genomics and Improvement Laboratory, USDA-ARS, Beltsville, MD, USA, using the Illumina Infinium SoySNP6K BeadChips (Illumina, Inc. San Diego, CA, USA) as previously described [53]. Subsequently, SNP alleles were called using GenomeStudio Genotyping Module 2.0 (Illumina, Inc. San Diego, CA, USA).

JoinMap 4.0 [54] was used to construct the genetic linkage map with a LOD score threshold of 3.0 and a maximum genetic distance of 50 cM to group markers. The linkage groups were assigned to corresponding soybean chromosomes as described in SoyBase [29,31].

### 3.4. Isoflavone QTL Detection and Statistical Analysis

The broad sense (mean based) heritability analysis from two-way ANOVA was conducted using the following equation: h^2^ = σ_G_^2^/[σ_G_^2^ + (σ_GE_^2^/e) + (σ_e_^2^/re)] where σ_G_^2^ (variance of genetic factor), σ_GE_^2^ (variance of genotype-environment interactions), and σ_e_^2^ (variance of random effect) were calculated with e (number of environment) and r (number of replicates) normalization [55]. R [56] was employed in the statistical analysis including agronomic traits, histogram of trait distribution, two-way ANOVA, and broad sense heritability using its native packages. The significant level of the assessed traits was showed using R package car (type II Wald chi-square tests) [56].

Both interval mapping (IM) and composite interval mapping (CIM) methods of WinQTL Cartographer 2.5 [39] were used to identify QTL for seed genistein, daidzein, glycitein, and total isoflavone contents in this RIL population. The default parameters of WinQTL Cartographer were chosen (Model 6, 1 cM step size, 10 cM window size, 5 control markers, and 1,000 permutations threshold) [39]. Chromosomes were drawn using MapChart 2.2 [57].

### 3.5. Isoflavone Candidate Genes Identification

The Glyma numbers of the isoflavone genes were obtained by searching the available data at the SoyBase [29,31] and Phytozome database [45]. The name of the isoflavone pathway enzymes (Figure 5) were used as a query in a search of the Glycine max reference genome, version Williams 82. The obtained isoflavone genes were mapped to the identified isoflavone QTL.

### 3.6. Expression Analysis

The expression analysis of the genes that are located within or near the isoflavone QTL was conducted using the publicly available soybean expression database from Phytozome database [45] to infer expression profiles of isoflavone genes in the soybean reference genome Williams 82. Gene expression was estimated in FPKM (Fragments Per Kilobase of transcript per Million mapped reads).

For the Forrest cv., RNA-seq library was prepared by using four plant soybean tissues including seed, leaf, root, flower and pods as shown earlier [58]. From 100 mg of frozen grounded samples, total RNA was extracted using RNeasy QIAGEN Kit (Qiagen, Hilden, Germany). The DNase I (Invitrogen, Carlsbad, CA, USA) was used to treat the total RNA. Using Illumina NovaSeq 6000, RNA-seq libraries preparation and sequencing were performed at Novogene INC. Multiplexing and sequencing of the four libraries were done in two different lanes generating 20 million raw pair end reads per sample (150 bp). Quality of sequenced reads was assessed using fastqc, version 0.11.9. [59]. The low-quality reads and adapters were removed with trimmomatic, version V0.39, the remaining high-quality reads were mapped to the soybean reference genome Wm82.a2.v1 using STAR, version v2.7.9 [60,61]. Uniquely mapped reads were counted using Python package HTseq v0.13.5. [62]. Read count normalization and differential gene expression analysis were conducted using the Deseq2 package v1.30.1 [63] integrated in the OmicsBox platform from BioBam (Valencia, Spain) [58,64].

## Figures and Tables

**Figure 1 plants-10-02029-f001:**
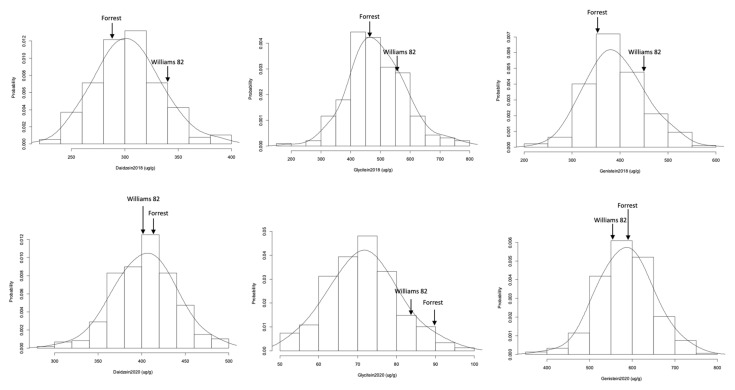
Frequency distribution of seed isoflavone contents (µg/g of seed weight) in the FxW82 RIL population. The seed daidzein, genistein, and glycitein contents were assessed in the RILs harvested in Spring Lake, NC (2018) and Carbondale, IL (2020).

**Figure 2 plants-10-02029-f002:**
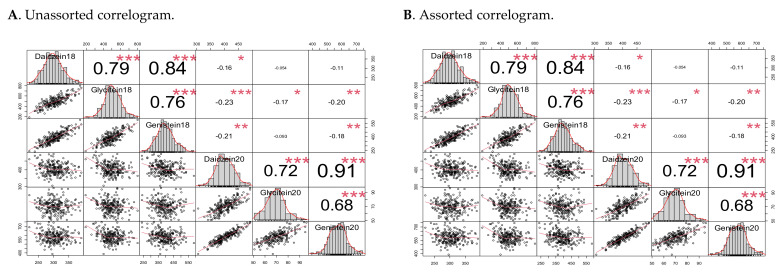
Correlations between daidzein, glycitein, and genistein in the two locations and years: Spring Lake, NC (2018) and Carbondale, IL (2020). (**A**). Unassorted correlogram, (**B**). Assorted correlogram. Significance level: * *p* < 0.05, ** *p* < 0.01, *** *p* < 0.001.

**Figure 3 plants-10-02029-f003:**
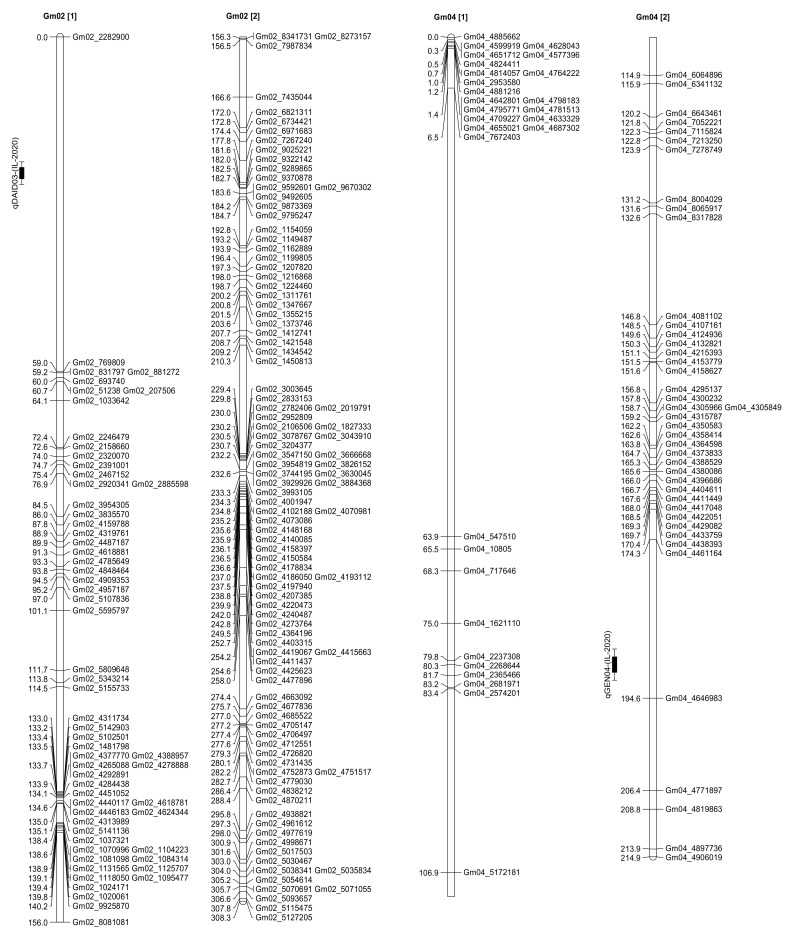
Positions of QTL that control seed genistein (qGEN), daidzein (qDAID), and glycitein (qGLY) contents on Chrs. 2, 4, 5, 6, 10, 12, 15, 19, and 20. QTL names are followed by a number, location, and year in which they are identified. For example, qGEN01-(NC-2018). The full SNP-based genetic linkage map of Forrest by Williams 82 recombinant inbred line (RIL) population (*n* = 306) of soybean is shown in Appendix A.

**Figure 4 plants-10-02029-f004:**
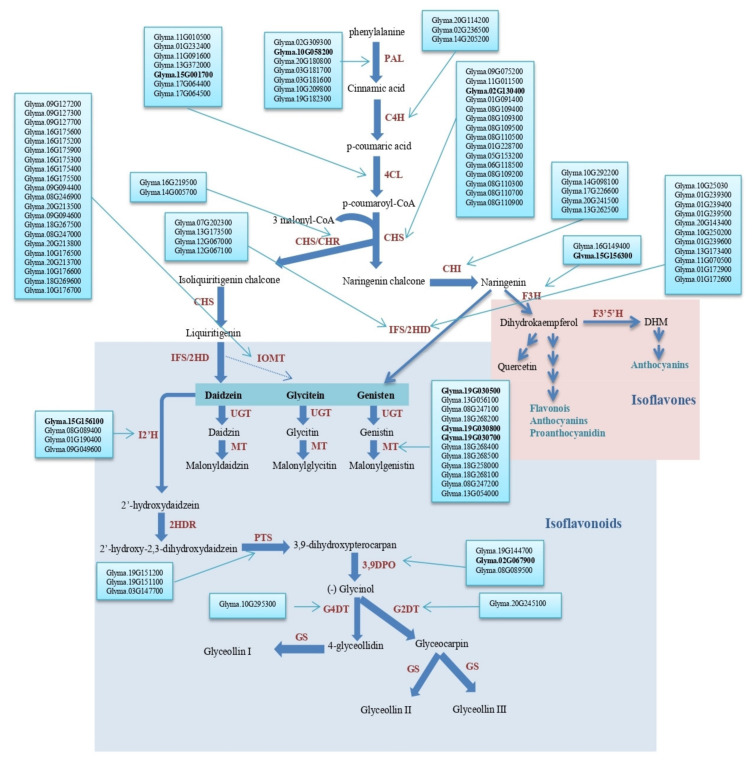
Seed isoflavone metabolic pathway in soybean with identified candidate genes (Vadivel et al., 2010). PAL, phenylalanine ammonia lyase; C4H, cinnamate 4-hydroxylate; 4CL, 4-coumarate-CoA ligase; CHS, chalcone synthase; CHR, chalcone reductase; CHI, chalcone isomerase; IFS, 2-hydroxyisoflavanone synthase; 2HID, 2-hydroxyisoflavanone dehydratase; IOMT, isoflavone O-methyltransferase; UGT, uridine diphosphate glycosyltransferase; MT, malonyltransferase; I2′H, Isoflavone 2′-hydroxylase; 2HDR, 2′-hydroxydaidzein reductase; F3H, flavanone-3-hydroxylase; F3′5′H, flavonoid 3′5′-hydroxylase; DHM, dihydromyricetin; PTS, pterocarpan synthase; 3,9 DPO, 3,9-dihydroxypterocarpan 6a-monooxygenase; G4DT, glycinol 4-dimethylallyltransferase; G2DT, glycinol 2-dimethylallyltransferase; GS, glyceollin synthase.

**Figure 5 plants-10-02029-f005:**
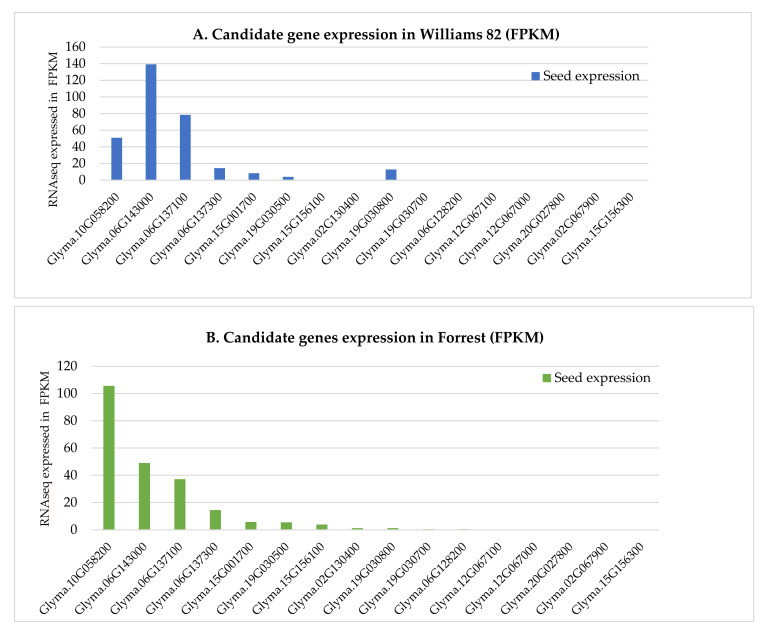
Expression pattern of isoflavone genes in soybean seeds. (**A**) Expression of the 16 isoflavone genes located within isoflavone QTL in Williams 82 (FPKM) were retrieved from publicly available RNA-seq data from Phytozome database [45], in addition to (**B**) the RNAseq data from the cultivar ‘Forrest’ (FPKM).

**Figure 6 plants-10-02029-f006:**
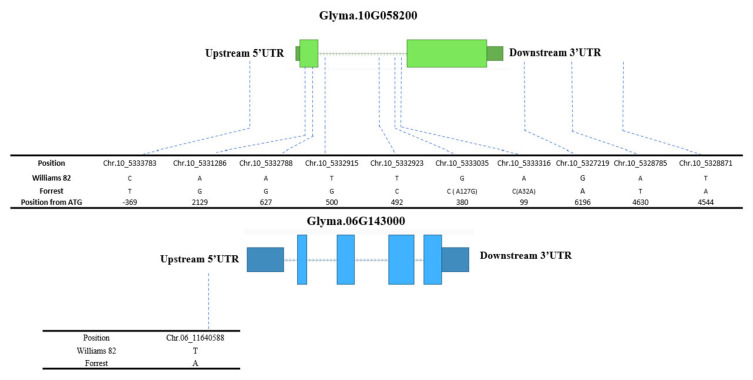
Positions of SNPs between Forrest and Williams 82 cultivars in Glyma.10G058200 and Glyma.06G143000 genes.

**Table 1 plants-10-02029-t001:** Distribution of SNP markers and their properties on the Chrs. Of Forrest by Williams 82 recombinant inbred line (RIL) population (*n* = 306).

Chr.	No. of SNP Markers	Length (cM)	Average Marker Density (cM)	Maximum Gap (cM)
1	110	190.1	1.73	48.7
2	161	308.3	1.91	59.0
3	92	173.9	1.89	22.9
4	71	214.9	3.03	57.4
5	138	167.2	1.21	38.1
6	114	253.7	2.23	43.4
7	117	224.0	1.91	18.4
8	71	211.1	2.97	45.3
9	109	179.2	1.64	62.6
10	100	216.5	2.17	48.5
11	95	168.9	1.78	41.3
12	73	192.6	2.64	31.5
13	156	265.7	1.70	57.7
14	50	158.9	3.18	25.8
15	94	219.1	2.33	68.4
16	95	169.0	1.78	46.7
17	79	185.4	2.35	46.5
18	144	153.7	1.07	23.1
19	125	190.4	1.52	53.3
20	81	187.3	2.31	25.7
Totals	2075	4029.9	Av. = 1.94	Av. = 43.2

**Table 2 plants-10-02029-t002:** Seed isoflavone means, ranges, CVs, skewness, and kurtosis in the FxW82 RIL population evaluated in Spring Lake, NC (2018) and Carbondale, IL (2020). Mean and range values are expressed in μg/g of seed weight.

Trait	Mean	Range	CV	SE	Skewness	Kurtosis	*p* Value (*p* > 0.05)
**Daidzein 2018**	303.22	171	10.11	2.23	0.26	3.11	0.99
**Glycitein2018**	490.47	610	19.37	7.01	0.29	3.65	0.98
**Genistein2018**	391	348	15.47	4.4	0.3	3.04	0.99
**Daidzein 2020**	14.48	8.08	13.72	0.42	−0.08	3.17	0.99
**Glycitein2020**	71.79	46	12.59	0.53	0.178	2.94	0.99
**Genistein2020**	584.88	383	10.94	3.73	−0.02	3.31	0.99

**Table 3 plants-10-02029-t003:** Two-way ANOVA results for daidzein, genistein, and glycitein.

Response: Daidzein				
	Df	Sum Sq	Mean Seq	H^2^
Line	301	541,800	1800	0.428
Year	1	974,711	974,711	
Line: Year	181	186,226	1029	
Residuals	0	0	NA	
**Response: Glycitein**				
	Df	Sum Sq	Mean Seq	H^2^
Line	301	5,086,274	16,898	0.724
Year	1	16,033,506	16,033,506	
Line: Year	181	843,473	4660	
Residuals	0	0	NA	
**Response: Genistein**				
	Df	Sum Sq	Mean Seq	H^2^
Line	301	1,922,339	6387	0.425
Year	1	3,630,207	3,630,207	
Line: Year	181	668,735	3695	
Residuals	0	0	NA	

**Table 4 plants-10-02029-t004:** QTL that control seed isoflavone (daidzein, genistein, and glycitein) contents in two environments over two years (2A. 2018 and 2B. 2020). The two environments are Spring Lake, NC (2018) (2A) and Carbondale, IL (2020) (2B). Only QTL with LOD scores > 2.0 and identified by composite interval mapping (CIM) method of QTL Cartographer (Wang et al., 2012) are reported.

2A. QTL Identified in Spring Lake, NC (2018)								
Trait	QTL	Chr.	Marker	Interval (cM)	LOD	R^2^ (%)	Additive Effect	Environment
Daidzein	*qDAID01*	5	Gm05_1705841	160.41	2.01	6.07	−7.61	Spring Lake, NC
Daidzein		5	Gm05_9012813	166.51	2.12	4.18	−6.35	Spring Lake, NC
Daidzein		5	Gm05_9097414	166.71	2.19	4.32	−6.46	Spring Lake, NC
Daidzein		5	Gm05_8916450	166.81	2.11	4.16	−6.34	Spring Lake, NC
Genistein	*qGEN03*	5	Gm05_1705841	152.41	2.06	9.37	−18.72	Spring Lake, NC
Genistein		5	Gm05_9012813	166.51	2.27	4.22	−12.53	Spring Lake, NC
Genistein		5	Gm05_9097414	166.71	2.36	4.39	−12.79	Spring Lake, NC
Genistein		5	Gm05_8916450	166.81	2.28	4.25	−12.57	Spring Lake, NC
Glycitein	*qGLY02*	5	Gm05_1705841	146.41	2.01	9.07	−29.89	Spring Lake, NC
Genistein	*qGEN01*	6	Gm06_5014399	64.41	2.58	8.52	21.15	Spring Lake, NC
Daidzein	*qDAID02*	6	Gm06_5014399	62.41	2.06	7.55	10.35	Spring Lake, NC
Daidzein		6	Gm06_3941524	78.21	2.02	7.24	8.67	Spring Lake, NC
Genistein	*qGEN04*	6	Gm06_5014399	60.41	2.26	9.06	22.94	Spring Lake, NC
Genistein		6	Gm06_3941524	70.21	2.11	3.98	13.86	Spring Lake, NC
Genistein	*qGEN02*	12	Gm12_915327	179.21	2.56	4.8	−16.87	Spring Lake, NC
Genistein		12	Gm12_1064727	179.41	2.58	4.85	−16.93	Spring Lake, NC
Genistein		12	Gm12_1229101	179.71	2.95	5.51	−17.79	Spring Lake, NC
Genistein		12	Gm12_1374970	179.91	2.95	5.5	−17.78	Spring Lake, NC
Genistein		12	Gm12_1433336	180.61	2.85	5.33	−17.4	Spring Lake, NC
Glycitein	*qGLY01*	12	Gm12_553862	177.31	2.76	5.82	−26.95	Spring Lake, NC
Glycitein		12	Gm12_915327	179.21	2.58	4.79	−27.05	Spring Lake, NC
Glycitein		12	Gm12_1064727	179.41	2.59	4.82	−27.12	Spring Lake, NC
Glycitein		12	Gm12_1229101	179.71	2.83	5.23	−27.76	Spring Lake, NC
Glycitein		12	Gm12_1374970	179.91	2.83	5.23	−27.76	Spring Lake, NC
Glycitein		12	Gm12_1433336	180.61	2.7	5.02	−27.03	Spring Lake, NC
Genistein	*qGEN05*	12	Gm12_553862	171.31	2.03	5.74	−15.18	Spring Lake, NC
Genistein		12	Gm12_975837	178.71	2.47	4.64	−16.47	Spring Lake, NC
Genistein		12	Gm12_1632399	181.31	2.32	4.38	−15.76	Spring Lake, NC
Glycitein	*qGLY03*	12	Gm12_553862	169.31	2.22	5.63	−23.71	Spring Lake, NC
Glycitein		12	Gm12_975837	178.71	2.46	4.58	−26.29	Spring Lake, NC
Glycitein		12	Gm12_1632399	181.31	2.2	4.1	−24.52	Spring Lake, NC
Daidzein	*qDAID03*	19	Gm19_4552537	109.51	2.14	4.17	6.92	Spring Lake, NC
Glycitein	*qGLY04*	19	Gm19_3010363	35.31	2.04	3.83	19.46	Spring Lake, NC
Genistein	*qGEN06*	20	Gm20_4657454	0.01	2.03	3.75	11.84	Spring Lake, NC
**2B. QTL Identified in Carbondale, IL (2020)**								
**Trait**	**QTL**	**Chr.**	**Marker**	**Interval (cM)**	**LOD**	**R2 (%)**	**Additive Effect**	**Environment**
Daidzein	*qDAID03*	2	Gm02_2282900	24.01	2.01	10.61	−11.55	Carbondale, IL
Genistein	*qGEN04*	4	Gm04_4461164	190.31	2.26	3.96	−12.97	Carbondale, IL
Genistein	*qGEN01*	10	Gm10_4670275	130.81	2.6	3.52	−12.38	Carbondale, IL
Genistein		10	Gm10_4035277	130.91	2.61	3.53	−12.39	Carbondale, IL
Daidzein	*qDAID04*	10	Gm10_4670275	130.81	2.18	2.97	−6.11	Carbondale, IL
Daidzein		10	Gm10_4035277	130.91	2.19	2.97	−6.12	Carbondale, IL
Genistein	*qGEN05*	10	Gm10_4670275	128.81	2.15	3.2	−11.75	Carbondale, IL
Genistein		10	Gm10_4035277	132.91	2.37	3.39	−12.11	Carbondale, IL
Daidzein	*qDAID01*	12	Gm12_9193994	53.21	2.56	4.07	7.18	Carbondale, IL
Daidzein		12	Gm12_1430950	61.71	3.99	5.61	8.49	Carbondale, IL
Daidzein		12	Gm12_1423120	62.31	4.12	5.78	8.62	Carbondale, IL
Daidzein		12	Gm12_1539402	63.01	4.53	6.34	9.01	Carbondale, IL
Daidzein		12	Gm12_1678702	63.11	4.59	6.45	9.1	Carbondale, IL
Daidzein		12	Gm12_3052701	64.11	4.89	6.82	9.42	Carbondale, IL
Daidzein		12	Gm12_2097199	64.41	4.4	6.18	8.97	Carbondale, IL
Daidzein		12	Gm12_2432082	65.31	4.26	5.97	8.77	Carbondale, IL
Daidzein		12	Gm12_1547239	65.51	3.77	5.31	8.27	Carbondale, IL
Daidzein		12	Gm12_1428801	65.91	3.36	4.75	7.87	Carbondale, IL
Genistein	*qGEN02*	12	Gm12_9193994	55.21	2.5	4.24	13.58	Carbondale, IL
Genistein		12	Gm12_1430950	61.71	3.78	5.28	15.31	Carbondale, IL
Genistein		12	Gm12_1423120	62.31	3.76	5.25	15.2	Carbondale, IL
Genistein		12	Gm12_1539402	63.01	4.39	6.1	16.4	Carbondale, IL
Genistein		12	Gm12_1678702	63.11	4.38	6.1	16.4	Carbondale, IL
Genistein		12	Gm12_3052701	64.11	4.66	6.46	16.99	Carbondale, IL
Genistein		12	Gm12_2097199	64.41	4.05	5.65	15.64	Carbondale, IL
Genistein		12	Gm12_2432082	65.31	3.83	5.35	15.25	Carbondale, IL
Genistein		12	Gm12_1547239	65.51	3.38	4.74	14.39	Carbondale, IL
Genistein		12	Gm12_1428801	65.91	3.16	4.44	13.96	Carbondale, IL
Daidzein	*qDAID05*	12	Gm12_9193994	51.21	2.03	2.91	6.05	Carbondale, IL
Daidzein		12	Gm12_1428801	73.91	2.31	4.8	7.83	Carbondale, IL
Genistein	*qGEN06*	12	Gm12_1428801	71.91	2.37	4.4	13.77	Carbondale, IL
Glycitein	*qGLY01*	15	Gm15_756303	212.31	2.07	2.86	−1.53	Carbondale, IL
Glycitein		15	Gm15_2072075	218.41	2.24	3.1	−1.6	Carbondale, IL
Glycitein		15	Gm15_2021199	218.81	2.06	2.84	−1.53	Carbondale, IL
Daidzein	*qDAID02*	20	Gm20_3804081	70.31	2.55	6.13	−8.81	Carbondale, IL
Genistein	*qGEN03*	20	Gm20_3804081	68.31	2.65	6.72	−16.81	Carbondale, IL
Daidzein	*qDAID06*	20	Gm20_3804081	66.31	2.22	5.58	−8.41	Carbondale, IL
Genistein	*qGEN07*	20	Gm20_3804081	64.31	2	5.08	−14.62	Carbondale, IL
Genistein		20	Gm20_3424023	80.01	2.44	3.3	−11.8	Carbondale, IL
Genistein		20	Gm20_3418121	80.51	2.1	2.85	−10.96	Carbondale, IL

**Table 5 plants-10-02029-t005:** Isoflavone candidate genes located within or close to the isoflavone QTL identified in the FxW82 RIL population two environments over two years (A. Spring Lake, NC (2018) and B. Carbondale, IL (2020)).

(A).							
Environment	Trait	QTL	Chr.	Gene	Start	End	Distance (cM)
2018 CIM QTL with LOD Scores > 2.5							
Spring Lake, NC	Genistein	*qGEN01*	6	Glyma.06G128200	10,543,911	10,545,747	5.52 cM
				Glyma.06G137100	11,225,188	11,228,664	6.21 cM
				Glyma.06G137300	11,237,072	11,239,469	6.22 cM
				Glyma.06G143000	11,642,031	11,644,022	6.62 cM
Spring Lake, NC	Genistein	*qGEN02*	12	Glyma.12G067000	4,909,073	4,911,905	3.47 cM
				Glyma.12G067100	4,919,960	4,922,998	3.48 cM
Spring Lake, NC	Glycitein	*qGLY01*	12	Glyma.12G067000	4,909,073	4,911,905	3.47 cM
				Glyma.12G067100	4,919,960	4,922,998	3.48 cM
**2018 CIM QTL with LOD Scores 2.0 < LOD < 2.5**							
Spring Lake, NC	Daidzein	*qDAID01*	5	-	-	-	-
Spring Lake, NC	Daidzein	*qDAID02*	6	Glyma.06G128200	10,543,911	10,545,747	5.52 cM
Spring Lake, NC	Daidzein	*qDAID03*	19	Glyma.19G030500	3,779,017	3,781,453	0.77 cM
				Glyma.19G030700	3,794,404	3,796,426	0.75 cM
				Glyma.19G030800	3,799,941	3,801,335	0.75 cM
Spring Lake, NC	Genistein	*qGEN03*	5	-			
Spring Lake, NC	Genistein	*qGEN04*	6	Glyma.06G128200	10,543,911	10,545,747	5.52 cM
Spring Lake, NC	Genistein	*qGEN05*	12	Glyma.12G067000	4,909,073	4,911,905	3.27 cM
				Glyma.12G067100	4,919,960	4,922,998	3.28 Cm
Spring Lake, NC	Genistein	*qGEN06*	20	Glyma.20G027800	3,179,955	3,183,453	1.47 cM
Spring Lake, NC	Glycitein	*qGLY02*	5	-	-	-	-
Spring Lake, NC	Glycitein	*qGLY03*	12	Glyma.12G067000	4,909,073	4,911,905	3.27 cM
				Glyma.12G067100	4,919,960	4,922,998	3.28 Cm
Spring Lake, NC	Glycitein	*qGLY04*	19	Glyma.19G030500	3,779,017	3,781,453	0.76 cM
				Glyma.19G030700	3,794,404	3,796,426	0.78 cM
				Glyma.19G030800	3,799,941	3,801,335	0.78 cM
**(B).**							
**Environment**	**Trait**	**QTL**	**Chr.**	**Gene**	**Start**	**End**	**Distance (cM)**
**2020 CIM QTL with LOD Scores > 2.5**							
Carbondale, IL	Daidzein	*qDAID01*	12	Glyma.12G067000	4,909,073	4,911,905	-
				Glyma.12G067100	4,919,960	4,922,998	-
Carbondale, IL	Daidzein	*qDAID02*	20	Glyma.20G027800	3,179,955	3,183,453	0.62 cM
Carbondale, IL	Genistein	*qGEN01*	10	Glyma.10G058200	5,328,963	5,333,501	0.6 cM
Carbondale, IL	Genistein	*qGEN02*	12	Glyma.12G067000	4,909,073	4,911,905	-
				Glyma.12G067100	4,919,960	4,922,998	-
Carbondale, IL	Genistein	*qGEN03*	20	Glyma.20G027800	3,179,955	3,183,453	2.44 cM
**2020 CIM QTL with LOD Scores 2.0 < LOD < 2.5**							
Carbondale, IL	Daidzein	*qDAID03*	2	Glyma.02G067900	5,986,285	5,987,684	3.70 cM
				Glyma.02G130400	13,399,253	13,401,493	11.11 cM
Carbondale, IL	Daidzein	*qDAID04*	10	Glyma.10G058200	5,328,963	5,333,501	0.6 cM
Carbondale, IL	Daidzein	*qDAID05*	12	Glyma.12G067000	4,909,073	4,911,905	-
				Glyma.12G067100	4,919,960	4,922,998	-
Carbondale, IL	Daidzein	*qDAID06*	20	Glyma.20G027800	3,179,955	3,183,453	-
Carbondale, IL	Genistein	*qGEN04*	4	-	-	-	-
Carbondale, IL	Genistein	*qGEN05*	10	Glyma.10G058200	5,328,963	5,333,501	0.6 cM
Carbondale, IL	Genistein	*qGEN06*	12	Glyma.12G067000	4,909,073	4,911,905	3.48 cM
				Glyma.12G067100	4,919,960	4,922,998	3.49 cM
Carbondale, IL	Genistein	*qGEN07*	20	Glyma.20G027800	3,179,955	3,183,453	2.44 cM
Carbondale, IL	Glycitein	*qGLY01*	15	Glyma.15G001700	190,985	194,451	0.56 cM
				Glyma.15G156300	13,098,492	13,100,036	11.02 cM
				Glyma.15G156100	13,076,997	13,079,333	11 cM

## Data Availability

Data supporting reported results are available on request from the corresponding author.

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
