# Peer review of "The Soybean High Density ‘Forrest’ by ‘Williams 82’ SNP-Based Genetic Linkage Map Identifies QTL and Candidate Genes for Seed Isoflavone Content"

_plants, 2021, doi:10.3390/plants10102029_

Round 1
Reviewer 1 Report
- The Results section
# comment.1
Table.2. must be further improved by adding more statistics. I would recommend to add information regarding number of RILs, minimum, maximum, and heritability values in the table and describe in the relevant section. Also, I would recommend to bring the results from ANOVA including the sources of variation (genotype and genotype-by-environment interaction). The analysis can be completed with phenotypic correlations between traits for each environment.
# comment.2
There is no section regarding the results from genetic linkage map construction of the populations. The authors must provide a section and explain the results before starting QTL analysis results. For example, total number of markers, final number of markers after filtration and quality assessment, total genetic distance and average distance between pairwise markers and genetic length of chromosomes, and the consistency of the markers with soybean genome. These must be carefully explained in the new section.
# comment.3
Following my previous comment, also I would suggest to bring a table explaining characteristics of high density map where the number of markers, map distance, average map distance, and maximum gap (cM) are indicated per each chromosome.
# comment.4
Figure 2. Is too huge, It can be shortened and only maintain some parts where major QTLs are located in. Others can be moved to supplementary information. Also it seems that there is no stable QTL detected across environments and the QTLs identified in this study have interaction effects. These results must be discussed and linked to the phenotypic analysis described in comment.1 above.
2. Materials and methods section
# comment.5
There is not sufficient information. How the materials were grown, the type of experimental design, number of replications, plant density including row distance and plant -to-plant distances must be explained.
# comment.6
All the analysis regarding (comment.1) described above must be explained in the MM section.
Author Response
We thank the reviewers for their valuable comments that we think will improve our manuscript. Below are our responses to the reviewers in red and the revised manuscript is attached.
Reviewer 1
Comments and Suggestions for Authors
- The Results section
# comment.1
Table.2. must be further improved by adding more statistics. I would recommend to add information regarding number of RILs, minimum, maximum, and heritability values in the table and describe in the relevant section. Also, I would recommend to bring the results from ANOVA including the sources of variation (genotype and genotype-by-environment interaction). The analysis can be completed with phenotypic correlations between traits for each environment.
Response: We updated the manuscript with this:
“We conducted broad sense (mean based) heritability analysis using two-way ANOVA based on following equation: h2=σG 2/[ σG 2 + (σGE 2/e) + (σe 2/re)] where σG 2 (variance of genetic factor), σGE 2 (variance of genotype-environment interaction), and σe 2 (variance of random effect) were calculated with e (number of environment) and r (number of replicates) normalization (Pilet-Nayel et al., 2002). The percentage of dry weight of Daidzein, Glycitein, and Genistein appeared to have quite different mean-based heritability of 42.8%, 72.4%, and 42.5%, respectively.”
# comment.2
There is no section regarding the results from genetic linkage map construction of the populations. The authors must provide a section and explain the results before starting QTL analysis results. For example, total number of markers, final number of markers after filtration and quality assessment, total genetic distance and average distance between pairwise markers and genetic length of chromosomes, and the consistency of the markers with soybean genome. These must be carefully explained in the new section.
Response: We do have section 2.1. The SNP-Based Genetic Map. The reviewer might overlook this section; however, the total number of SNP markers (5,405) and final number of markers after filtration (2,075), total genetic distance (4,029.9 cM), and average distance between markers (1.94 cM), and genetic length of chromosomes (153.7 cM for Chr. 18 to 308.3 cM for Chr. 2 and details in Table 1) were mentioned. However, this section is updated with more details.
# comment.3
Following my previous comment, also I would suggest to bring a table explaining characteristics of high density map where the number of markers, map distance, average map distance, and maximum gap (cM) are indicated per each chromosome.
Response: This information is provided in Table 1. Maximum gap information for each chromosome is added in a new column added to this Table.
# comment.4
Figure 2. Is too huge, It can be shortened and only maintain some parts where major QTLs are located in. Others can be moved to supplementary information. Also it seems that there is no stable QTL detected across environments and the QTLs identified in this study have interaction effects. These results must be discussed and linked to the phenotypic analysis described in comment.1 above.
Response: Figure 2 is split into Fig. 2. that is included in the manuscript with only the chromosomes containing the QTL and Fig. S1. that contains all chromosomes is moved the Supplementary Tables and Figures.
- Materials and methods section
# comment.5
There is not sufficient information. How the materials were grown, the type of experimental design, number of replications, plant density including row distance and plant -to-plant distances must be explained.
Response: The section 3.1. Plant Material and Growth Conditions was updated with more details as suggested by both reviewers.
# comment.6
All the analysis regarding (comment.1) described above must be explained in the MM section.
Response: This has been addressed in Materials and Methods as suggested by the reviewer.
Sincerely,
MA Kassem.

Reviewer 2 Report
The study provides comprehensive insight into genetics underlying important traits for isoflavone contents in soybean cultivars. The aim was to construct a SNP-based genetic linkage map using the F×W82 RIL population and to map quantitative trait loci (QTL) for seed daidzein, genistein, glycitein, and total isoflavone contents in two environments over two years. A total of 27 QTL controlling different seed isoflavone trait have been identified and mapped on various chromosomes. Among the 27 mapped QTLs, 6 were not identified by previous studies. The authors also detected 130 candidate genes involved in isoflavone biosynthetic pathways of soybean. In addition they carried out RNA-Seq analysis in F x W82 (RIL) population for 16 candidate genes located within or near the identified isoflavone QTLs. This analysis detected 4 genes with high expression in seeds of both Forrest and Williams 82 cultivars. The subject matter falls within the scope of "Plants" journal. The paper shows an original contribution in the area of soybean genetics and breeding. The QTL methodology is appropriate and the text flow is well organized although it is a bit descriptive.
Hereafter, some specific comments for the authors to consider.
- There were quite a number of markers which could not be linked to any other marker. A total of 5,405 SNP markers were generated but only 2075 were mapped. This is unexpected in a progeny of this size and given the total number of markers, which is enough to get a very dense map. This may point to large number of missing values (technical problems encountered during analysis) or considerable genotypic errors. No explanation is provided.
- There is strong evidence that below ground competition is an important structuring force in crop communities and often at least equally intense as above ground competition. Resource competition below ground is largely based on exploitation (i.e., nutrient availability, climate etc,). The authors do not provide details about the design of the plantation (i.e density) and which fertilizers were used during the cultivation for seed production in 2018 & 2020. It is important to know the growing conditions of the plants in order to interpret the isoflavone levels response of the identified QTLs.
- Line 103. What type of unit the values represent (μ g/mg? ) in the isoflavone data for soybean samples (table 2)? I see that the values of daidzein, glycitein, in 2020 are extremely low compared to 2018 and much below from the standard values in food industry. But this is not the case for genistein. Does this reflect different growing conditions between 2018 && 2020? The authors should explain about this inconsistency.
- The authors should provide more technical details on the quantification of the isoflavones. The reference provided is not explaining much. For example, what spectra used in NIR diode array feed analyzer. Which model of analyzer was used? Did they perform any calibration before? Any control experiment? How many scans per sample were run? etc
- In my opinion an interesting part of the work is the assessment of the isoflavone contents traits in 2 years in 2 environments. This provides the possibility to the authors to make valuable statement about QTL stability over years or environments. Nothing is really discussed on this. It is an important outcome concerning molecular breeding or MA selection strategies. In general, I think the paper is very descriptive and lacks discussion.
- Line 351, "genotypic data from 119 RILs have been deleted from QTL analysis. The authors should clarify why this large portion of data ( 119 RILs) excluded from QTL analysis.
- What does the Y axis represent in figure 4? It is not clear.
- The number of QTLs shown in figure 2 are 26 and not 27 as stated in the text. The authors should correct this.
- The authors should provide in the very beginning the ploidy of the soybean cultivars crossed. This may be helpful for readers outside the research field of soybean.
- I would prefer the "conclusion part" before the M & Methods.
Author Response
We thank the reviewers for their valuable comments that we think will improve our manuscript. Below are our responses to the reviewers in red and the revised manuscript is attached.
Reviewer 2
Comments and Suggestions for Authors
The study provides comprehensive insight into genetics underlying important traits for isoflavone contents in soybean cultivars. The aim was to construct a SNP-based genetic linkage map using the F×W82 RIL population and to map quantitative trait loci (QTL) for seed daidzein, genistein, glycitein, and total isoflavone contents in two environments over two years. A total of 27 QTL controlling different seed isoflavone trait have been identified and mapped on various chromosomes. Among the 27 mapped QTLs, 6 were not identified by previous studies. The authors also detected 130 candidate genes involved in isoflavone biosynthetic pathways of soybean. In addition they carried out RNA-Seq analysis in F x W82 (RIL) population for 16 candidate genes located within or near the identified isoflavone QTLs. This analysis detected 4 genes with high expression in seeds of both Forrest and Williams 82 cultivars. The subject matter falls within the scope of "Plants" journal. The paper shows an original contribution in the area of soybean genetics and breeding. The QTL methodology is appropriate and the text flow is well organized although it is a bit descriptive.
Hereafter, some specific comments for the authors to consider.
- There were quite a number of markers which could not be linked to any other marker. A total of 5,405 SNP markers were generated but only 2075 were mapped. This is unexpected in a progeny of this size and given the total number of markers, which is enough to get a very dense map. This may point to large number of missing values (technical problems encountered during analysis) or considerable genotypic errors. No explanation is provided.
Response: We added this information on the polymorphism of SNPs in this RIL:
The polymorphism of SNPs in this RIL population (38.4%), number of linked SNPs, and map coverage were comparable to other reported SNP-based genetic linkage maps of soybean [61-62]. For example, in Akond et al. (2013) [61], only 27.33% of SNPs (1,465/5,361x100) have been used to construct the genetic map based on excluding missing data (~20%) and heterozygosity (3.99%). Polymorphic markers between parents (MD96-5722 and Spencer) among the 1,465 SNPs used was 44.8% (657/1,465x100) [61]. Likewise, in Kassem et al. (2012) [62], polymorphic markers between parents (PI 438489B and Hamilton) among the 1,465 SNPs used was 44.2% (657/1,465x100) [62].
- There is strong evidence that below ground competition is an important structuring force in crop communities and often at least equally intense as above ground competition. Resource competition below ground is largely based on exploitation (i.e., nutrient availability, climate etc,). The authors do not provide details about the design of the plantation (i.e density) and which fertilizers were used during the cultivation for seed production in 2018 & 2020. It is important to know the growing conditions of the plants in order to interpret the isoflavone levels response of the identified QTLs.
Response: The section 3.1. Plant Material and Growth Conditions was updated with more details as suggested.
- Line 103. What type of unit the values represent (μg/mg? ) in the isoflavone data for soybean samples (table 2)? I see that the values of daidzein, glycitein, in 2020 are extremely low compared to 2018 and much below from the standard values in food industry. But this is not the case for genistein. Does this reflect different growing conditions between 2018 && 2020? The authors should explain about this inconsistency.
Response: This was μg/g of seed weigh. The manuscript is updated with this:
Azam et al. (2020) [50] reported that the total isoflavones ranged from 745 μg/g to 5,253.98 μg/g, with highest mean of 2689.27 μg/g was observed in some regions and upto 2,518.91 μg/g and 1,942.78 μg/g in others due to climatic conditions. Similar results have been reported by other studies [51-56]. Our results showed over 1,000 μg/g and in some cases over 1,100 μg/g. Therefore, the total concentrations of isoflavones are in the expected range of soybean seed. In addition, there is no premium to be given to growers for soybean seed isoflavone content and no docking is done at the grain elevator for seed isoflavon. Isoflavone concentrations vary depending on the year and environmental growing conditions. Although isoflavone are genetically controlled, environmental conditions including temperature, drought, absence or presence of diseases each year, many other biotic and abiotic factors can significantly affect the contents (by increasing or decreasing) and profile of isoflavone.
- The authors should provide more technical details on the quantification of the isoflavones. The reference provided is not explaining much. For example, what spectra used in NIR diode array feed analyzer. Which model of analyzer was used? Did they perform any calibration before? Any control experiment? How many scans per sample were run? Etc.
Response: This information is now added to the section “3.2. Isoflavone Quantification”.
- In my opinion an interesting part of the work is the assessment of the isoflavone contents traits in 2 years in 2 environments. This provides the possibility to the authors to make valuable statement about QTL stability over years or environments. Nothing is really discussed on this. It is an important outcome concerning molecular breeding or MA selection strategies. In general, I think the paper is very descriptive and lacks discussion.
Response: We added some parts concerning this in discussion.
- Line 351, "genotypic data from 119 RILs have been deleted from QTL analysis. The authors should clarify why this large portion of data ( 119 RILs) excluded from QTL analysis.
Response: This statement was added “In September 2018, hurricane Florence hit NC and its winds of >90 mph damaged the fence in the farm in Spring Lake, NC; therefore, the deers damaged about 119 RILs. The plants grown in Carbondale, IL (n=304) were not damaged; therefore, data from the 119 RILs has been excluded in QTL analysis for results uniformity.”
- What does the Y axis represent in figure 4? It is not clear.
Response: This was added: Gene expression was estimated in FPKM (Fragments Per Kilobase of transcript per Million mapped reads).
- The number of QTLs shown in figure 2 are 26 and not 27 as stated in the text. The authors should correct this.
Response: There are 27 QTL in Fig. 2. as stated in text and Table 3. We verified all of them. Maybe the reviewer needs to pay attention to the 5 QTL on Chr. 20.
- The authors should provide in the very beginning the ploidy of the soybean cultivars crossed. This may be helpful for readers outside the research field of soybean.
Response: We added this statement “The genomes of soybean cultivars including Forrest and Williams 82 genomes are duplicated polyploid genomes with highly conserved gene-rich regions [49, Shultz et al., 2006].”. Reference [41] was added to the references section.
- I would prefer the "conclusion part" before the M & Methods.
Response: It is now before Materials and Methods as suggested.
Sincerely,
MA Kassem

Round 2
Reviewer 1 Report
The authors has improved the manuscript. However beside parts of my previous comments which are still missing, there are few important points that must be clarified to be understandable and followed by reader.
# comment-1
A table of Anova containing all components of variance and heritability values must be provided for each trait. The correlogram showing pairwise correlations between traits for each environment to know the degree of relationship between studied traits.
# Comment-2
The heritability value for Glycation is relatively high enough (72.4%) to detect QTL with stable effect. What could be the reason ? It would be nice to bring a table showing the properties of environments where the material were grown. For example temperature, humidity, soil properties….
# Comment -3
Perhaps the most important part is section 2.3 describing results of QTL analysis. It must be reorganized for example, chromosome by chromosome or trait by trait. Or by single trait QTL and QTL with pleiotropic effects. The current version doesn’t show any particular structure and can be hardly followed by reader.
# Comment -4
It is not clear to me why the available data from 119 lines were excluded for Carbondale, IL? As the authors performed QTL analysis separately for each environment, I don’t think excluding lines with available data causing any problem and in contrast, it would improve the precision of QTL analysis. Therefore I highly recommend to not exclude any data.
Author Response
We appreciate your comments that really improved the manuscript. We addressed the comments below to the best of our ability.
# comment-1
A table of ANOVA containing all components of variance and heritability values must be provided for each trait. The correlogram showing pairwise correlations between traits for each environment to know the degree of relationship between studied traits.
Response: Table 3, Figure 2 are now provided. Two paragraphs (lines 134-145 and 151-152 address this comment). Thank you.
# Comment-2
The heritability value for Glycitein is relatively high enough (72.4%) to detect QTL with stable effect. What could be the reason ? It would be nice to bring a table showing the properties of environments where the material were grown. For example temperature, humidity, soil properties….
Response: We provided this information in the manuscript’s text (lines 400-412): “In Spring Lake, NC (2018) during the growing season (May–Sept.), the temperatures ranged from 7.20C to 350C, it was partly (40%) to mostly cloudy (80%), wind speeds ranged from 55 mph to 90+ mph (hurricane Florence), and humidity comfort level ranged from comfortable to miser-able (weatherspark.com). In Carbondale, IL (2020), the temperatures ranged from 7.20C to 29.40C, it was mostly clear (25%) to mostly cloudy (80%), wind speeds ranged from 30 mph to 38 mph, and humidity comfort level ranged from comfortable to miserable (weatherspark.com). The field was treated first using Firestorm (contains Paraquat dichloride) to control annual grass and broad-leaved weeds. As pre-emergent herbicide, Dual II Magnum Herbicide with long-lasting control of most annual grasses and small-seeded broadleaf weeds was used to eliminate early-season weed competition. As post-emergent herbicide, Round Up Pro Concentrate (50.2% Glyphosate) was used/sprayed between the rows to control emerging weed. Weed grown inside the plastic mulch very close to the soybeans were removed manually”.
The soil is sandy in Spring Lake, NC (NC Sandhills) and silty clay loam in Carbondale, IL. Glycitein high heritability is mostly due to the environmental and growth conditions. Explanation is provided in text.
# Comment -3
Perhaps the most important part is section 2.3 describing results of QTL analysis. It must be reorganized for example, chromosome by chromosome or trait by trait. Or by single trait QTL and QTL with pleiotropic effects. The current version doesn’t show any particular structure and can be hardly followed by reader.
Response: This section is now organized chromosome by chromosome following Table 4 as suggested by the reviewer (see lines 164-240). Table 4 was also rearranged.
# Comment -4
It is not clear to me why the available data from 119 lines were excluded for Carbondale, IL? As the authors performed QTL analysis separately for each environment, I don’t think excluding lines with available data causing any problem and in contrast, it would improve the precision of QTL analysis. Therefore, I highly recommend to not exclude any data.
Response: This was a mistake from the corresponding author (MAK). The QTL analysis did not exclude any data. Both 2018 and 2020 data sets were analyzed independently without excluding any data points. Originally, MAK asked JY to exclude these data points from QTL analysis for results uniformity; however, JY did not exclude the data points and MAK assumed he did. The manuscript text is updated, and QTL Cartographer’s QTL data analysis files are available upon request. We are sorry for the confusion.
Sincerely,
MA Kassem.
Reviewer 2 Report
The authors have addressed the concerns raised and the present version is now ready for publication.
Author Response
Thank you very much!